# Therapeutic Potential of Two Derivative Prescriptions of Rokumijiogan, Hachimijiogan and Bakumijiogan against Renal Damage in Nephrectomized Rats

**DOI:** 10.3390/medicines10030024

**Published:** 2023-03-21

**Authors:** Chan Hum Park, Takashi Tanaka, Yoshie Akimoto, Jin Pyeong Jeon, Takako Yokozawa

**Affiliations:** 1Institute of New Frontier Research Team, Hallym Clinical and Translational Science Institute, Hallym University, Chuncheon 24252, Republic of Korea; 2Graduate School of Biomedical Sciences, Nagasaki University, Nagasaki 852-8521, Japan; 3Iskra Industry Co., Ltd., Tokyo 103-0027, Japan; 4Department of Neurosurgery, College of Medicine, Hallym University, Chuncheon 24252, Republic of Korea; 5Graduate School of Science and Engineering for Research, University of Toyama, Toyama 930-8555, Japan

**Keywords:** nephrectomy, Rokumijiogan, Hachimijiogan, Bakumijiogan, renal damage

## Abstract

**Background:** Hachimijiogan (HJG) and Bakumijiogan (BJG), two derivative prescriptions of Rokumijiogan (RJG), were selected to investigate their renoprotective potential in the 5/6 nephrectomized (5/6Nx) rat model. **Methods:** Rats were treated with HJG and BJG orally at 150 mg/kg body weight/day once daily for 10 weeks after resection of 5/6 of the renal volume, and their renoprotective effects were compared with 5/6Nx vehicle-treated and sham-operated control rats. **Results:** Improvements in renal lesions, glomerulosclerosis, tubulointerstitial injury, and arteriosclerotic lesions estimated by histologic scoring indices in the HJG-treated group were compared with those in the BJG-treated group. HJG- and BJG-treated groups ameliorated the renal function parameters. Elevated levels of renal oxidative stress-related biomarkers were reduced, while decreased antioxidant defence systems (superoxide dismutase and the glutathione/oxidized glutathione ratio) were increased in the HJG-treated group rather than the BJG-treated group. In contrast, BJG administration significantly reduced expression of the inflammatory response through oxidative stress. The HJG-treated group showed a decrease in inflammatory mediators through the JNK pathway. To gain a deeper understanding of their therapeutic action, the effects of the main components detected in HJG and BJG were evaluated using the LLC-PK_1_ renal tubular epithelial cell line, which is the renal tissue most vulnerable to oxidative stress. Corni Fructus and Moutan Cortex-originated compositions afforded important protection against oxidative stress induced by peroxynitrite. **Conclusions:** From our described and discussed analyses, it can be concluded that RJG-containing prescriptions, HJG and BJG are an excellent medicine for chronic kidney disease. In the future, appropriately designed clinical studies in people with chronic kidney disease are necessary to evaluate the renoprotective activities of HJG and BJG.

## 1. Introduction

Chronic kidney disease (CKD) is one of the major public health problems worldwide. Over the past 20 years, the number of deaths from CKD has increased by 82.3%, which is approximately the third largest increase among the 25 leading causes of death [1]. Treatment of stages 1 to 4 of patients with CKD focus on slowing kidney damage and preventing or treating complications and comorbidities through medication or diet. In the final stage (stage 5 or end-stage renal disease) of CKD, renal replacement therapy such as dialysis or kidney transplantation is needed to sustain life [2]. The prognosis of CKD patients has improved with the development of kidney dialysis and medical management systems. However, maintaining continuous dialysis is a clear burden on patients in terms of both physical and mental parts, and social issues including financial burdens have arisen due to the overall number of patients receiving dialysis. Several conservative treatments for CKD may be applied in this situation.

Herbal medicines, including traditional medicines using natural products, have attracted a lot of attention due to their significant and varied pharmacologic actions and bioactive phytoconstituents without side effects or toxicity [3,4]. Medicinal plants, also called medicinal herbs, have been used against various diseases for thousands of years around the world. For example, medicinal plant species with antioxidant properties including phenolics, carotenoids, and vitamins are widely engaged as traditional medicine and functional food and help prevent and/or treat diseases caused from an overproduction of reactive oxygen species (ROS) such as cancer, neurodegenerative diseases, and cardiovascular disease [5]. According to the viewpoint of traditional Chinese medicine (TCM), CKD is commonly considered a Qi (vital energy) deficiency and Yang (warms and activates the body) shortage, that is, exhausted vigor caused by Xieqi (evil influence) repleteness. The basis of clinical management is to replenish Yangqi and drain Xieqi [6]. Herbal drug compositions containing Rokumijiogan (RJG) are used for renal insufficiency in traditional East Asian medicine. It is one of the representative improving body function medications and is widely used to treat frequent urination, micturition disorder, and edema, resulting from nephrosis and chronic nephritis [4,7,8].

Among RJG-containing prescriptions, we became interested in the Hachimijiogan (HJG) and Bakumijiogan (BJG), which may be useful in treating patients with renal disease. HJG is composed of RJG, Cinnamomi Cortex, and Aconiti Tuber. BJG is also composed of RJG contained in HJG. Six crude drugs contained in RJG and two additional crude drugs, Ophiopogonis Tuber and Schisandrae Fructus, constitute BJG (Table 1). HJG has long been widely used to treat multiple chronic conditions and diseases, including diabetes, vegetative ataxia, infertility, and nephritis, although scientific and pharmacological evidence supporting its therapeutic effectiveness has not been well demonstrated. Especially, it has been widely used to treat impaired renal function in human research subjects with hypertension, glomerulonephritis, and nephritic disorders [9,10,11]. BJG has been used to treat renal impairment based on its antioxidant activity [12]. However, there is little experimental data on its use as a prophylactic or therapeutic drug.

Two derivative prescriptions (HJG and BJG) of RJG and their components have been reported to have renoprotective effects against CKD. These two derivative prescriptions of RJG are anticipated to provide a novel therapeutic approach to nephrectomy. However, the renoprotective mechanism of HJG and BJG had not been known until compounds [7-*O*-galloyl-D-sedoheptulose (GS), pentagalloyl glucose (PG), paeonol, cornuside, and schizandrin] originating from Corni Fructus, Moutan Cortex, and Schisandrae fructus were analyzed as the major active components of HJG and BJG. Thus, this review aims to describe the therapeutic effects of HJG and BJG against CKD, along with the underlying mechanisms of such renoprotective effects and bioactive components. Therefore, we investigated whether HJG and BJG improved oxidative stress-triggered inflammatory conditions associated with nephrectomy. Our data suggest that HJG and BJG may be novel therapeutics for improving CKD [13,14]. To gain a deeper understanding of their therapeutic action, the effects of the main crude drug components on renal cells needed to be evaluated. We chose a renal proximal tubular epithelial cell line, because this type of cell is one of the main targets of attack by oxygen free radicals [15,16].

## 2. HJG and BJG Extracts

The ingredients of HJG and BJG, which form a traditional Chinese herbal formula, are shown in Table 1. HJG is comprised of the following (the values after the name indicate the proportion of each ingredient expressed in parts per whole): Radix Rehmanniae 6.0, Fructus Corni 3.0, Rhizoma Dioscoreae 3.0, Rhizoma Alismatis 3.0, Poria 3.0, Cortex Moutan 2.5, Cortex Cinnamomi 1.0, and Radix Aconiti 0.5. These eight crude drugs were boiled gently in 10 times their volume of water for 60 min, filtered, and the powder was spray-dried to obtain an extract at a yield of about 10% by weight of the original preparation. The composition of BJG was: Radix Rehmanniae 0.750, Fructus Corni 0.370, Rhizoma Dioscoreae 0.370, Rhizoma Alismatis 0.280, Poria 0.280, Cortex Moutan 0.280, Fructus Schisandrae 0.199, and Radix Ophiopogonis 0.280. The BJG powder was prepared with a yield of about 12% by weight of the crude drug. To analyze the components, HJG and BJG were separately pulverized and a portion (0.5 g) was extracted by vortexing at room temperature for 8 h with 5 mL of MeOH and then sonicated for 20 min. After filtration through a 0.45 mm membrane filter, an aliquot (10 μL) of the sample solution was injected into the high-performance liquid chromatography (HPLC) column. Chromatographic separations were performed on a Cosmosil 5C_18_-AR II (Nacalai Tesque, Kyoto, Japan) reverse-phase column (250 × 4.6 mm i.d.), using 50 mM H_3_PO_4_ (A) and CH_3_CN (B) as the mobile phase at 4 to 30% (39 min) and 30 to 75% (15 min). The column temperature was 40 °C and flow rate was 0.8 mL/min. The separated compounds were detected by JASCO MD-2010 HPLC diode array detectors, identified, and quantified on the basis of chromatographic retention times of coeluted pure standards using Class LC-10 Version 1.62 software (Shimadzu, Kyoto, Japan). Eleven compounds (GS, gallic acid, morroniside, loganin, paeoniflorin, PG, cornuside, cinamic acid, 6′-*O*-benzoyl paeoniflorin (BP), cinamdalehyde, and paeonol) of HJG extract were detected in the HPLC chromatogram. In the elution profile of BJG extract, GS, gallic acid, morroniside, loganin, paeoniflorin, PG, cornuside, BP, paeonol, schizandrin, gomisin A, and schizandrin were identified [13].

## 3. Renal Histological Findings and Renal Functional Parameters

Prolonged proteinuria due to subtotal nephrectomy causes irreversible structural changes in nephrons and renal dysfunction, including glomerular-capillary hypertension, proximal tubule damage, stimulation of nuclear signals such as nuclear factor-kappa B (NF-κB), and NF-κB-mediated inflammatory events in the interstitium, interstitial inflammatory responses, and consequently, fibrosis [17,18]. Moreover, an enlarged glomerular function resulting from nephrectomy is elicited by high glomerular filtration, and glomerular enlargement and destruction, eventually bringing about glomerulosclerosis [19]. These observations indicate that renal ablation induced the early stage of CKD with renal morphologic changes, such as accelerated mesangial matrix production, mesangial cell proliferation, and glomerular endothelial cell injury, as well as tubulointerstitial damage. In our histopathological study, clear histological types indicating glomerular sclerosis, tubulointerstitial damage, and arteriolar sclerotic lesion were discovered after nephrectomy, whereas glomerulo and tubulointerstitial damage was ameliorated in 5/6 nephrectomized (5/6Nx) rats that were orally administered HJG and BJG. Arteriolar sclerotic lesions were also improved [13]. Although improvement in activity on the progression of kidney lesions was more marked in the HJG-treated group than in the BJG-administered group, the two types of Chinese medicine prescriptions showed improvement in renal pathological lesions by nephrectomy. Uremic toxins maintained due to renal dysfunction by direct or secondary effects on renal tissue aggravate renal function and tissue, generating a vicious cycle that results in end-stage renal disease [20]. Consistent with this, rats given HJG and BJG showed improvements in biochemical parameters including renal profiles (urinary protein, blood urea nitrogen, serum creatinine, and creatinine clearance) [13].

## 4. Biomarkers Associated with Oxidative Stress in the Kidney

In experiments using unilaterally 5/6Nx rats, decreased antioxidative activity and increased oxidative stress in the remaining kidney were observed by Harris et al. [21], Ozbek [22], Tamay-Cach et al. [23], and Ling and Kuo [24]. Enhanced ROS production leads to various forms of tissue damage and functional disabilities by attacking, denaturing, and changing structural and functional molecules and by activating redox-responsive transcriptional factors and the signal transduction pathway. These phenomena sequentially increase necrotic and apoptotic cell death, inflammatory-driven fibrosis, and other injuries. In the kidney, oxidative stress is a persistent hallmark of progressive renal disease, and it plays an important role in progressive worsening of the renal function and various structures, as well as numerous other complications [25]. Therefore, we initially investigated the effects of HJG and BJG on oxidative stress-related items associated with the development of kidney disease using 5/6Nx rats.

We found that renal ROS levels in vehicle-treated 5/6Nx rats were significantly higher than those of sham control rats, but in the Nx-BJG group, the ROS levels showed that a clear sign of recovery almost sham control levels after administration of BJG for 10 weeks. However, there were no differences between the vehicle-treated 5/6Nx and HJG-treated 5/6Nx groups. Moreover, levels of renal thiobarbituric acid-reactive substances (TBARS) were 0.52 nmol/mg protein in the sham control group, whereas it was significantly increased in the 5/6Nx-Veh group (1.19 nmol/mg protein). These levels of renal TBARS were markedly decreased in BJG-treated 5/6Nx rats. Conversely, the activities of the renal antioxidant enzymes (superoxide dismutase, reduced glutathione/oxidized glutathione ratio) in the renal tissue in 5/6Nx-Veh rats were significantly decreased compared with sham control rats (*p* < 0.001), whereas both HJG- and BJG-treated groups showed the effect of increasing the activity of antioxidant enzymes compared to those of the 5/6Nx-Veh group. Unlike the consequence of oxidative metabolism, the activities of antioxidant enzymes showed greater in the 5/6Nx-HJG groups than in the 5/6Nx-BJG groups [14]. These results suggest that HJG or BJG administration can be considered as having different mechanisms.

## 5. Renal NADPH Oxidase-4 (Nox-4), p22^phox^, Nrf2, and Heme Oxygenase-1 (HO-1) Protein Expressions

Although the causes of increased ROS generation in kidney disease are very diverse, our studies have focused on the fact that NADPH oxidase mainly contributes to the process of ROS generation [26]. Structurally, NADPH oxidase includes a membrane-associated cytochrome *b*_558_ consisting of one p22^phox^ (phox is an acronym for phagocyte oxidase), one gp91^phox^ subunit, and at least four cytosolic components (p47^phox^, p67^phox^, p40^phox^, and the small GTPase rac1 or rac2) [27]. Notably, among them, Nox-4 and p22^phox^ have been identified as major sources of ROS production in tissues and cells and could play roles in some pathological conditions [28,29,30]. Therefore, we evaluated the protein expression of the NADPH oxidase subunits Nox-4 and p22^phox^ in the renal tissue to establish the precise mechanism in HJG- and BJG-administered rats. Chemiluminescent Western blot data indicate that the expressions of both Nox-4 and p22^phox^ were upregulated significantly in the kidney of 5/6Nx rats. On the contrary, BJG-treated rats showed a significant reduction in these protein expressions, whereas administration of HJG significantly reduced only Nox-4 expression. Especially, the inhibitory activates on the expression of two NADPH oxidase units were strong in BJG-treated rats [13]

In addition, oxidative stress provokes variations in the Nrf2 complex and the upregulated transcription of Nrf2-dependent genes such as HO-1 [31]. Nrf2 is a redox-sensitive transcription factor involved in protecting cells from oxidative stress-related damage [32,33]. NADPH oxidase-derived ROS and ROS-mediated mitogen-activated protein kinases (MAPKs) may arbitrate the induction of antioxidant HO-1 and its transcription factor, Nrf2 [34,35]. Therefore, regulating the Nrf2/HO-1 antioxidant signaling pathway may be a promising target of the pathology of oxidative stress. In our results, the 5/6Nx vehicle-treated group showed reduced expressions of the Nrf2/HO-1 antioxidant signaling pathway in the remnant renal tissue compared to those of sham control rats. Otherwise, administration of BJG effectively relieved oxidative stress and upregulated Nrf2 and HO-1 [13]. Activation of intracellular signaling cascades, including NADPH oxidase-derived ROS and, consequently, induced MAPKs may arbitrate the stimulation of Nrf2/HO-1 expression.

## 6. Renal c-Jun N-Terminal Kinase (JNK), Phosphor (p)-JNK, c-Jun, and Transforming Growth Factor-β_1_ (TGF-β_1_) Protein Expressions

JNK activation is triggered by proinflammatory cytokines and ROS. JNK has been suggested to play a critical role in signaling events via the activation of activator protein-1 transcription factor, phosphorylation of c-Jun, and induction of proinflammatory molecules expression such as intercellular adhesion molecule-1 (ICAM-1). ROS generation is also induced by JNK activation [36]. JNK activation can be modulated via stabilization/gene expression through its phosphorylation, p-JNK [37]. Only nuclear-translocated p-JNK induces c-Jun upregulation [38]. TGF-β-induced fibronectin expression requires JNK activation, which, in turn, regulates c-Jun activation [39]. This being so, we demonstrated that the JNK-dependent TGF-β1 signaling pathway was significantly increased in vehicle-treated 5/6Nx rats compared with that in sham-operated rats. On the contrary, HJG-treated rats showed significantly reduced expression of these proteins in the kidney [13].

## 7. Renal NF-κB, Cyclooxygenase-2 (COX-2), Inducible Nitric Oxide Synthase (iNOS), Monocyte Chemotactic Protein-1 (MCP-1), and ICAM-1 Protein Expressions

NF-κB signaling is a central pathway of inflammatory stimuli involved in the modulation of transcription and multiple inflammation-related mediator expressions such as COX-2 and iNOS [40]. The severe and progressive renal damage in the 5/6Nx model was associated with increased NF-κB activity, whereas the amelioration of proteinuria and renal structural damage was associated with the inhibition of NF-κB system activity. Increased oxidative stress induces the expression of MCP-1 and ICAM-1 genes. Macrophage infiltration regulated by chemokines such as MCP-1, and adhesion molecules such as ICAM-1, has been reported to be related with deposition of renal immune complex and elevated production of renal chemokine [41,42]. As shown in the study published by Park et al. [13,14], it showed that renal NF-κBp65 and its associated inflammation-related mediator, chemokine, and adhesion molecule expressions were markedly increased in vehicle-treated 5/6Nx rats compared with sham-operated rats. In contrast, the BJG-treated group showed significant suppression of the expression of NF-κBp65 and NF-κB-mediated target proteins. ICAM-1 protein expression was also reduced by BJG administration. The administration of HJG led to significant down-regulation of NF-κBp65 and COX-2 expression, whereas it slightly altered iNOS and MCP-1 levels. Therefore, our results suggest that NF-κB and NF-κB target-inflammatory genes such as inflammatory mediators, chemokines, and adhesion molecules are important factors in the renoprotective activity of BJG.

## 8. Protective Role of RJG-Containing Components against Free Radical-Induced Oxidative Stress in Renal Tubular Epithelial Cells

Reactive oxygen and free radicals have various effects on the progression and development of renal disease. The results of several studies have provided evidence that renal tubular obstruction is involved in renal damage, and it is well known that renal tubular cells, including LLC-PK_1_ renal proximal tubule cells, are susceptible to free radical damage [43,44]. Therefore, an in vitro model of oxidative damage, in which LLC-PK_1_ cells are exposed to free radicals, would be useful in the search for compounds that can provide effective protection.

As mentioned above [13], GS, gallic acid, morroniside, loganin, paeoniflorin, PG, cornuside, BS, and paeonol were identified as major components of HJG and BJG extracts. In addition, schizandrin was detected in the BJG extract. Therefore, we evaluated the protective effects of these ten compounds in the LLC-PK_1_ renal tubular epithelial cell line, which is the renal tissue most vulnerable to oxidative stress.

Although various kinds of free radical initiators are known, 3-morpholinosydnonimine (SIN-1) has been used successfully as a peroxynitrite initiator, since it generates free radicals at a measurable and constant rate via thermal decomposition without biotransformation. To elucidate the protective effect against SIN-1 in cells, we measured cell viability and demonstrated that the exposure of renal tubular LLC-PK_1_ cells to SIN-1 resulted in decreased cell viability. Among the 10 components tested in this study, the most potent activity was exhibited by GS, which showed the highest viability (89.4% at 50 μg/mL) and protected against cell damage induced by SIN-1 in time- and dose-dependent manners. PG, paeonol, and cornuside also had inhibitory effects, suggesting that the compounds originating from Corni Fructus and Moutan Cortex protected LLC-PK_1_ cells from SIN-1-induced cytotoxicity. Schizandrin, originating from Schisandrae Fructus and contained in BJG, presented similarly strong viability of 67.4% at 50 μg/mL. These findings suggest that Corni Fructus- and Moutan Cortex-originating compositions might act as free radical scavengers and afford important protection against oxidative stress induced by SIN-1.

## 9. Conclusions

According to the underlying concepts of TCM theory, CKD is commonly regarded as caused by Qi deficiency and Yang deficiency due to Xieqi repletion. The basic treatment principle of TCM is Yang/Qi-replenishing and Xieqi-draining. HJG and BJG, two derivative prescriptions of RJG used in this studym included both aspects of TCM theory: complementary and extirpated medicinal herbs. To compare the renoprotective effects and mechanisms of HJG and BJG prescriptions, we focused on oxidative stress-associated factors related with the development of CKD, which was advanced in 5/6Nx rats. The two RJG-containing prescriptions played protective roles against the progression of CKD. HJG treatment of 5/6Nx rats prevented CKD progression primarily through amelioration of inflammation through inhibition of JNK signaling in the kidney, whereas BJG exerted a protective effect against regulation of NADPH oxidase and Nrf2/HO-1 signaling. Based on these results in 5/6Nx rats, we suggest the existence of multiple pathways by which RJG-containing prescriptions protected against the development of CKD associated with nephrectomy. In addition, compounds (GS, PG, paeonol, cornuside, and schizandrin) originating from Corni Fructus, Moutan Cortex, and Schisandrae fructus may play important roles in strengthening the protective effects against cell damage. Therefore, this study may be concluded that two RJG-containing prescriptions (HJG and BJG) may prevent the functional disorder and morphological destruction of the kidney that is associated with the progression of CKD and may improve the redox balance and inflammatory response in the kidney.

## Figures and Tables

**Table 1 medicines-10-00024-t001:** Compositions of HJG and BJG.

Pharmaceutical Name	Scientific Name	HJG	BJG
Radix Rehmanniae Preparata	*Rehmannia glutinosa* LIB. var. *purpurea* MAKINO	✓	✓
Rhizoma Dioscoreae	*Dioscorea japonica* THUNB.	✓	✓
Fructus Corni	*Cornus officinalis* SIEB. et ZUCC.	✓	✓
Hoelen	*Poria cocos* WOLF	✓	✓
Rhizoma Alismatis	*Alisma plantago-aquatica* L. subsp. *orientale* SAMUELSSON	✓	✓
Cortex Cinnamomi	*Cinnamomum cassia* BLUME	✓	✓
Cortex Moutan Radicis	*Paeonia suffruticosa* ANDREWS	✓	-
Radix Aconiti Lateralis Preparata	*Aconitum carmichaeli* DEBX.	✓	-
Fructus Schisandrae	*Schisandra chinensis* BAILLON	-	✓
Raidix Ophiopogonis	*Ophiopogon japonicus* KER-GAWLER var. genuinus MAXIM.	-	✓

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
