# Peer review of "Therapeutic Potential of Two Derivative Prescriptions of Rokumijiogan, Hachimijiogan and Bakumijiogan against Renal Damage in Nephrectomized Rats"

_medicines, 2023, doi:10.3390/medicines10030024_

Round 1

Reviewer 1 Report

The current manuscript concerns the potential of several plant extracts for treatment of renal damage. It is an extensive review on their potential, and is reasonably well written. Nevertheless, some alterations should be made before publication:

- Methods and results should be more summarized in the abstract; furthermore, a final “conclusion” sentence in missing from the abstract;

- Section 2 is written as if the authors themselves performed the analysis; this should be corrected, and the appropriate references should be added; same with figure 1: where was it taken from? Please provide the appropriate reference, and the permission for reusing the image must also be provided, and written in the figure’s caption;

- Overall, it should be better stated in the text where the commented data comes from; some sections lack references, some have references but should start the sentence with “In a study by Author et al.”, otherwise it becomes more confusing to read;

- Line 133 has an incomplete sentence, “final stage of CKD [17].”; this should be corrected;

- All figure captions and tables should have the reference of the publication where they come from, and the authors should provide a permission to use statement in all captions; such as “reproduced from author et al. (licence number XYZ)”;

- Where does Figure 6 come from? Did the authors make it themselves? If so, the used software should be mentioned, and the image should be provided as a whole, without being in “parts”;

- There are many abbreviations in the text, and hence an abbreviation list needs to be added.

Author Response

Dear Editor,

Manuscript ID: medicines-2150783

Title: Therapeutic Potential of Two Derivative Prescriptions of Rokumi-jio-gan, Hachimi-jio-gan and Bakumi-jio-gan, against Renal Damage in Nephrectomized Rats

Authors: Chan Hum Park, Takashi Tanaka, Yoshie Akimoto, Jin Pyeong Jeon, Takako Yokozawa

We are grateful to the editor and reviewers for the critical comments and useful suggestions that have helped us to improve our paper considerably. We have studied the reports carefully, and have revised the manuscript accordingly (the revised parts are highlighted). The responses to the comments are as follows:

Reviewer #1

Comment: The current manuscript concerns the potential of several plant extracts for treatment of renal damage. It is an extensive review on their potential, and is reasonably well written. Nevertheless, some alterations should be made before publication:

Comment: Methods and results should be more summarized in the abstract; furthermore, a final “conclusion” sentence in missing from the abstract;

Response: The abstract has been more summarized.

Commemt: Section 2 is written as if the authors themselves performed the analysis; this should be corrected, and the appropriate references should be added; same with figure 1: where was it taken from? Please provide the appropriate reference, and the permission for reusing the image must also be provided, and written in the figure’s caption;

Response: The reference has been added in the caption of Figure 1.

Comment: Overall, it should be better stated in the text where the commented data comes from; some sections lack references, some have references but should start the sentence with “In a study by Author et al.”, otherwise it becomes more confusing to read;

Response: The reference haws been added and described in lines line 119-121, line 256-257.

Comment: Line 133 has an incomplete sentence, “final stage of CKD [17].”; this should be corrected;

Response: It has been revised (line 159).

Comment: All figure captions and tables should have the reference of the publication where they come from, and the authors should provide a permission to use statement in all captions; such as “reproduced from author et al. (licence number XYZ)”;

Response: It has been added.

Comment: Where does Figure 6 come from? Did the authors make it themselves? If so, the used software should be mentioned, and the image should be provided as a whole, without being in “parts”;

Response: We made it ourselves (before revision: Figure 6; after revision: graphical abstract). The image in Figure 6 is the whole file, not parts.

Comment: There are many abbreviations in the text, and hence an abbreviation list needs to be added.

Response; According to the comment, abbreviation list has been added in the last part of abstract.

Yours sincerely,

Dr. Chan Hum Park

Prof. Takako Yokozawa

Reviewer 2 Report

The abstract should be better summarized.

Lines 43-48, these sentences should be enlarged.

The use of natural products in traditional medicine and the importance of research network should be better marked and related references should be added such as:

Singla, et al. (2023). The International Natural Product Sciences Taskforce (INPST) and the power of Twitter networking exemplified through #INPST hashtag analysis. Phytomedicine : international journal of phytotherapy and phytopharmacology108, 154520. https://doi.org/10.1016/j.phymed.2022.154520

The novelty character of this review respect to other ones in literature should be better marked.

At the end of Introduction, the authors should better describe and introduce the different paragraphs and better link them.

In all Figures and Tables the authors should specify the sources and the references of the data and results reported.

I suggest to not report a Figure in Conclusion.

Author Response

Dear Editor,

Manuscript ID: medicines-2150783

Title: Therapeutic Potential of Two Derivative Prescriptions of Rokumi-jio-gan, Hachimi-jio-gan and Bakumi-jio-gan, against Renal Damage in Nephrectomized Rats

Authors: Chan Hum Park, Takashi Tanaka, Yoshie Akimoto, Jin Pyeong Jeon, Takako Yokozawa

We are grateful to the editor and reviewers for the critical comments and useful suggestions that have helped us to improve our paper considerably. We have studied the reports carefully, and have revised the manuscript accordingly (the revised parts are highlighted). The responses to the comments are as follows:

The comments were valuable and helpful to revise the manuscript. We believe that the revised manuscript, being sent herewith, is a marked improvement. Therefore, we hope that it is now acceptable for publication in medicines. Thank you for your consideration.

Yours sincerely,

Dr. Chan Hum Park

Prof. Takako Yokozawa
